# Exploring the Geroprotective Potential of Nutraceuticals

**DOI:** 10.3390/nu16172835

**Published:** 2024-08-24

**Authors:** Nadia Alejandra Rivero-Segura, Emmanuel Alejandro Zepeda-Arzate, Selma Karime Castillo-Vazquez, Patrick Fleischmann-delaParra, Jessica Hernández-Pineda, Edgar Flores-Soto, Paola García-delaTorre, Edgar Antonio Estrella-Parra, Juan Carlos Gomez-Verjan

**Affiliations:** 1Dirección de Investigación, Instituto Nacional de Geriatría (INGER), Mexico City 10200, Mexico; nrivero@inger.gob.mx (N.A.R.-S.); bm.ezepeda@gmail.com (E.A.Z.-A.); selmakarime@gmail.com (S.K.C.-V.); patrickfp15@gmail.com (P.F.-d.); 2Posgrado en Ciencias Biológicas, Universidad Nacional Autónoma de México, Mexico City 04510, Mexico; 3Departamento de Infectología e Inmunología, Instituto Nacional de Perinatología, SSA, Mexico City 11000, Mexico; jesspinq@yahoo.com.mx; 4Departamento de Farmacología, Facultad de Medicina, Universidad Nacional Autónoma de México, Avenida Universidad No. 3000, Alcaldía de Coyoacán, Mexico City 04510, Mexico; edgarfloressoto@yahoo.com.mx; 5Unidad de Investigación Epidemiológica y en Servicios de Salud, Área Envejecimiento, Centro Médico Nacional Siglo XXI, Instituto Mexicano del Seguro Social, Mexico City 06720, Mexico; pgarciatorre@gmail.com; 6Laboratorio de Fitoquímica, UBIPRO, FES-Iztacala, Universidad Nacional Autónoma de México, Tlalnepantla de Baz 54090, Mexico; estreparr@iztacala.unam.mx

**Keywords:** nutraceuticals, geroprotectors, aging, hallmarks of aging, bioactive compounds

## Abstract

Aging is the result of the accumulation of a wide variety of molecular and cellular damages over time, meaning that “the more damage we accumulate, the higher the possibility to develop age-related diseases”. Therefore, to reduce the incidence of such diseases and improve human health, it becomes important to find ways to combat such damage. In this sense, geroprotectors have been suggested as molecules that could slow down or prevent age-related diseases. On the other hand, nutraceuticals are another set of compounds that align with the need to prevent diseases and promote health since they are biologically active molecules (occurring naturally in food) that, apart from having a nutritional role, have preventive properties, such as antioxidant, anti-inflammatory and antitumoral, just to mention a few. Therefore, in the present review using the specialized databases Scopus and PubMed we collected information from articles published from 2010 to 2023 in order to describe the role of nutraceuticals during the aging process and, given their role in targeting the hallmarks of aging, we suggest that they are potential geroprotectors that could be consumed as part of our regular diet or administered additionally as nutritional supplements.

## 1. Introduction

Food is the source of energy required for all physiological functions and biological activities, as well as the main source of all essential compounds that cannot be naturally synthesized by our organism [1]. Moreover, the effect of food on our health depends on the balance of our diet and on how efficiently and effectively our organism utilizes nutrients (carbohydrates, lipids, proteins, vitamins and minerals) [2]. Interestingly, a healthy life and, consequently, a long lifespan requires balanced nutrition. In this sense, the availability, amount and quality of food varies depending on factors such as origins, composition and manufacturing process, among others. 

Nutrients are traditionally classified according to the function they play in our organism. Therefore, nutrients that contribute energy, such as lipids, proteins and carbohydrates, are known as macronutrients. While trace amounts of vitamins, minerals and other organic compounds, which do not directly contribute to energy metabolism, are known as micronutrients or bioactive compounds [3]. In the past decades, micronutrients have been linked to overall well-being and to health benefits associated with disease prevention and treatment. These compounds were termed “nutraceuticals” by Stephen DeFelice in 1995 by joining the concept of pharmaceutical bioactivity with that of nutrients found in our foods [4]. Interestingly, such a term, consequently, has been cited indifferently in the literature, leading to a varied number of definitions that oftentimes are contradictory. Additionally, nutraceuticals’ definition tends to be confused with the term “functional foods”, which describes specific types of food that are characterized by a high content of bioactive compounds with relevant effects on well-being and health or in reducing the risk of diseases [5]. Interestingly, to date, there is no consensus nor a unique definition for “nutraceuticals” and/or “functional foods”. For the current article, we use the term nutraceutical as biologically active molecules naturally occurring in food that, apart from having a nutritional role, provide health-promoting, disease-curing or prevention properties [6]. A nutraceutical must be understood as a single substance that may be isolated for clinical purposes or consumed as part of a specific food [7].

On the other hand, aging as a process affects different levels of the biological hierarchy and significantly affects molecular pathways that, once altered, are associated with several types of diseases, such as cardiovascular, neurodegenerative, cancer, metabolic disorders and many other syndromes [8]. In this context, current aging research focuses on developing strategies to slow down the detrimental effects of aging. In this sense, over the past years the term “geroprotector” has become significant as potential molecules that target the so-called Hallmarks of Aging [9], delaying the onset of age-related diseases and boosting resilience in older adults [10]. The so-called “geroprotectors” stand out, named as such by Illya Mechnikov, one of the fathers of gerontology, who defined them as an agent that allows protection against the effects of aging, thus increasing life expectancy and healthy life span [11]. 

Interestingly, some of the current geroprotector molecules have been discovered by repurposing previously approved drugs or already tested compounds (most of which are derived from natural sources consumed in the diet). Recently, the development of different-omics tools has allowed redefining the term by a unifying concept in a more formal way as that intervention that delays, reduces and/or prevents diseases associated with aging and that is characterized by having a simultaneous target of one or several of the pillars of aging [12]. Therefore, in the present review, we summarize the activities of nutraceuticals that target the molecular hallmarks of aging that could be consumed as part of our regular diet or administered additionally as nutritional supplements, suggesting the design of a potential “geroprotective diet” that could be defined as a diet that could reduce or prevent adverse outcomes of aging and that may contain a bioactive compound that targets one or several of the hallmarks of aging.

For the selection of the studies summarized in this review, we performed research by using the boolean operators AND/OR, NOT and Medical Subject Headings (MeSH) words, such as “nutraceutics”, “nutraceuticals”, “functional food”, “aging”, “geroprotectors” and each of the hallmarks of aging (“telomere attrition”, “epigenetics”, “loss of proteostasis”, “disabled macroautophagy”, “deregulated nutrient sensing”, “mitochondrial dysfunction”, “cellular senescence”, “stem cell exhaustion”, “altered intercellular communication”, “chronic inflammation”, “dysbiosis” and “genomic instability”). We only consider original articles published from 2010 to 2023 in Scopus and Pubmed databases and exclude papers that were in other languages different from English or without data published. A total of 69 articles were included to perform the following review, and the nutraceutical classification was performed according to [13]. 

## 2. Nutraceuticals

In essence, from a chemical perspective, classic nutraceutical compounds are considered small chemical structures naturally synthesized in the secondary metabolism of plants. This classification includes characteristic chemical structures, such as anthraquinones, alkaloids, saponins, tannins, essential oils, carotenoids, flavonoids and bitters [6]. However, in recent years, carbohydrates, protein and lipids have also been introduced as nutraceuticals together with other essential micronutrients. Lastly, probiotics, prebiotics and fungal extracts have also been partially considered nutraceuticals due not only to their direct effect as such, but mainly because of their ability to metabolize compounds into more bioactive molecules [14]. For instance, as depicted in Figure 1 most of the common nutraceuticals are found in daily foods.

The field of nutraceuticals is still too young compared to those of pharmaceuticals, and there is still a lack of rigorous large-scale clinical trials. Nevertheless, over the last 10 years, there has been significant growth in scientific evidence that suggests that they could be helpful for the treatment of cardiovascular and metabolic diseases, cognitive function, skin physiology, immune function and digestive health [15]. In this sense, the nutraceuticals market is experiencing significant growth due to the increased focus on preventative healthcare, awareness about the benefits of dietary supplements and the growing geriatric population accompanied with rising age-related disease prevalence. It is estimated that the global nutraceutical market (functional food, dietary supplements and functional beverages) is about 554.7 billion USD and will grow to 905.8 billion USD by 2030 [16]. Accordingly, Table 1 provides a summary of selected studies on foods, bioactive compounds or nutraceuticals and their main targets and functions investigated in models of aging or aging-related diseases.

## 3. Nutraceuticals as Geroprotectors

According to Illya Mechnikov, geroprotectors are agents that protect against the adverse effects of aging, thus increasing life expectancy and healthy lifespan [11]. Moreover, the current advances in gerosciences help us to understand biological mechanisms that are closely associated with aging, also known as the Hallmarks of Aging (genomic instability, telomere attrition, epigenetic alterations, loss of proteostasis, deregulated nutrient sensing, mitochondrial dysfunction, cellular senescence, stem cell exhaustion and altered intercellular communication, disabled macroautophagy, chronic inflammation and dysbiosis) [9]. Each hallmark contributes to the aging process and together determines the aging phenotype while meeting a number of criteria: (i) it should manifest during normal aging; (ii) its experimental aggravation should accelerate aging; and (iii) its experimental amelioration should retard the normal aging process and, hence, increase healthy lifespan [49]. Altogether, with this information, we can redefine geroprotector in a more formal way as an intervention targeting simultaneously one or more of the hallmarks of aging delaying, reducing and/or preventing age-related diseases [12]. Hence, in the following section, we aim to suggest the potential properties of nutraceuticals as geroprotectors by summarizing the most outstanding studies where nutraceuticals target the hallmarks of aging. 

### 3.1. Telomere Attrition

Telomeres are the caps at the ends of chromosomes that protect them from fraying and fusing with other chromosomes or becoming damaged by an external agent. In this sense, telomere attrition is an interesting hallmark of aging since telomeres become shorter in each cell division till it is so short that it generates a DNA damage signaling response. In this context, telomerase (a reverse transcriptase enzyme) adds bases to the ends of telomeres and is responsible for telomere length. In most adult cells, telomerase activity is low, leading to progressive telomere shortening with each division till telomeres grow shorter and DNA becomes damaged, conditioning the fate of cells to senescence or death [50]. Several studies show a correlation between shorter telomere length and increased risk of age-related diseases. Hence, it has been proposed that certain compounds, including polyphenols, triterpenes, sesquiterpenes, xanthones and alkaloids derived from natural products have the potential to modulate telomerase activity, suggesting their possible use as anti-aging agents [51]. 

It has been reported that a three-month administration of vitamin C, zinc and vitamin D3 results in a reduction in the rate of telomere shortening and an increase in telomere length, suggesting that these vitamins may promote telomerase activity [52]. It is noteworthy that the acid form at a low concentration of vitamin A (retinoic acid) reduces p16^INK4A^ expression and induces telomerase activity, increasing the lifespan of human oral keratinocytes [53]. Moreover, both in vitro and in vivo studies, as well as epidemiological studies, demonstrate that the use of vitamin C, vitamin D and vitamin E are associated with individuals with longer telomere lengths than their counterparts [54]. 

Interestingly, a one-year randomized, double-blind, placebo-controlled study performed in older adults infected with cytomegalovirus and administered with TA-65 (a compound derived from the herb *Astragalus membranaceus* [55]) demonstrated a significant increase in telomere length [56]. Other studies demonstrated that polyphenols, such as curcumin and resveratrol, obtained from grapes, cherries and blueberries, among others, are being studied for their potential anti-aging effects [51]. For instance, the administration of resveratrol to aged C57Bl/6 mice over a one-year period resulted in a decrease in telomere attrition. Notably, this treatment has been observed to have a differential effect according to sex, restoring the length of telomeres significantly in brain cells of aged female mice but not in males [57]. Omega-3 fatty acids (EPA, DHA and polyunsaturated fatty acids (PUFAs)) have been demonstrated to possess anti-inflammatory properties, which are associated with the maintenance of telomere length [55]. A randomized pilot study of older adults with mild cognitive impairment who received supplementation with linolenic acid (LA), EPA or DHA for six months demonstrated that telomere shortening was attenuated. Furthermore, elevated levels of DHA in erythrocytes were associated with decreased telomere shortening [58]. 

Although telomerase is a promising therapeutic target, overexpression of the enzyme occurs in approximately 90% of cancers, which makes telomerase-related anti-aging therapies somewhat controversial due to potential adverse effects [59]. Therefore, the results of compounds interacting with telomerase should be taken with caution, and research overcoming such limitations are currently needed. 

### 3.2. Epigenetics

Epigenetic changes influence the aging process in a number of ways, including reduced levels of core histone marks, changes in the patterns of histone post-translational modifications, DNA methylation and altered expression of non-coding RNAs (ncRNAs). These changes can trigger aberrant gene expression and genomic instability [60]. Nevertheless, epigenetic changes can also act transgenerationally, influencing the lifespan of the offspring, as demonstrated in studies performed in *C. elegans*, where deficiencies of any of the chromatin modifier components result in lifespan extension for up to three generations [61].

Given that epigenetic modifications are reversible, the use of nutraceuticals represents an attractive strategy for regulating epigenetically active enzymes or the epigenome itself [62]. Indeed, it has been demonstrated that polyphenols, flavonoids and organosulfur compounds in foods (vitamin A, vitamin C, vitamin E, curcumin and resveratrol) exert epi-nutraceutical effects, modifying DNA methylation patterns, histone modifications and regulating miRNA expression [63]. Although the exact effect mechanisms by which vitamins interact with the DNA methylation process is yet to be investigated, there are some experiments that suggest that vitamin A administered in vitro to human embryonic stem cells (hESCs), induced hypermethylation of most genes and a decrease in the H3K27me3 repressive mark [64]. Additionally, it has been reported that vitamin E enhances the expression of DNMT1 and LINE-1. On the other hand, vitamin C seems to activate TET hydroxylase enzymes. Similarly, curcumin has been shown to inhibit DNMT1 and DNMT3B, as well as to inhibit HATs, HDAC2 and HDAC8 [65]. Moreover, curcumin upregulates the expression of miR-15, -16, -9 and -181 and downregulates expression of miR-125b-5p, -19a, -19b, -27a and -130a [66]. Finally, resveratrol was also shown in vitro on HCC1806 breast cancer cells to inhibit DNMT enzymatic activity and to modulate HDAC activity in different models both in vivo and in vitro [67].

Retinoic acid is a micronutrient that modifies one-carbon metabolism; consequently, deficiencies in such micronutrients result in decreased DNA methylation due to the availability of methyl groups [68]. A study involving older adults reported that multivitamin supplementation with vitamins B3, C and D, omega-3 fish oils (EPA and DHA), resveratrol, olive phenols and astaxanthin for 12 weeks demonstrates that older adults with an initial epigenetic age acceleration of ≥2 years at baseline showed a significant reduction in both epigenetic age and its acceleration after the supplementation [69]. In addition, curcumin and resveratrol, along with other polyphenols, have been reported to activate sirtuin 1 (SIRT1), which deacetylates histones, leading to the regulation of various biological functions, including metabolism, cellular senescence, inflammatory processes and stress, as it deacetylates p53, forkhead transcription factor, NF-κB and the LX receptors [70,71].

### 3.3. Loss of Proteostasis

The coordinated action of proteostasis networks, including chaperones, the ubiquitin-proteasome system (UPS) and the lysosome-autophagy system, enables proteostasis by affecting the key processes of protein synthesis, translation regulation, protein folding and protein clearance [72,73]. These networks detect and rectify alterations in the proteome; with aging, the maintenance of proteostasis is compromised in cells and organs under resting and stress conditions [72].

Chaperone proteins, such as heat shock proteins (Hsp), can be modulated by certain nutraceuticals, including curcumin and proanthocyanidins present in cranberry extract [74]. An in vitro model of rat glioma C6 cells treated with curcumin (3–10 μM) and exposed to arsenite, cadmium chloride or heat (42 °C for 30 min) showed the synthesis of Hsp27 and Hsp70 proteins [75]. Interestingly, Hsp27 is responsible for protein degradation and controls apoptosis by regulating Akt activation, while Hsp70 has functions in protein folding and unfolding [76,77]. This mechanism of action of Hsp has been studied for the removal of peptide plaques associated with neurodegenerative diseases that occur in the aging population, such as Alzheimer’s disease (AD) [76]. It is noteworthy that the proanthocyanidins present in cranberry extract promote proteostasis, in addition to improving the lifespan of *C. elegans* in an AD model. In this model, an IIS-dependent increase in HSF-1 activity has been observed, resulting in the reduction in β-amyloid (Aβ) peptide species and increased protein solubility in old worms [78].

The UPS can be activated through nutraceuticals, such as linolenic acid present in dairy products, and olein, which can be consumed through palm oil; these compounds can exert conformational changes that favor the entry of substrates through the proteolytic chamber [74,79]. Oleuropein, a compound derived from olives and olive oil, has been observed to stimulate proteasome activity in vitro on human embryonic fibroblasts; as a result, the lifespan of these cells was extended by the 20S proteasome alpha channels [80]. Similarly, resveratrol has been examined in AD transgenic mice (3xTg-AD) supplemented from two to ten months old with resveratrol, resulting in an increase in neprilysin, an enzyme responsible for amyloid degradation. Additionally, there was an increase in the levels of the central subunits of the 20S proteasome, as well as an increase in the concentration of Hsp70 and ubiquitinated proteins (improving proteostasis activity) [81]. The study of nutraceuticals that maintain or promote proteostasis is not only beneficial for the study of aging but also for the prevention and treatment of diseases related to age-related protein aggregation [74].

### 3.4. Disabled Macroautophagy

A key protective mechanism of aging is the cellular recycling process called autophagy. During autophagy, damaged cellular components are delivered to acidic vesicles called lysosomes, which secure the degradation and recycling of the components [82]. As we age, autophagy becomes less efficient and contributes to the aging process and the concomitant development of age-related diseases. Particularly, macroautophagy is induced in response to different stressors, such as nutrient or growth factor deprivation, hypoxia, damaged proteins and organelles and genotoxic stress, among others. This response is tightly regulated by a variety of signaling pathways, including mTOR and AMPK, to secure appropriate fine-tuning [83]. While research on manipulating autophagy for longevity benefits in humans is still ongoing, the potential of autophagy-based interventions for promoting healthy aging and preventing diseases is a very promising area of research.

In this sense, the use of quercetin and curcumin induces autophagy via downregulated mTOR, P53 and P21 protein expression and decreased the phosphorylation of AKT, mTOR and P70S6K, respectively, on a model of atherosclerosis and melanoma [84,85]. Studies showed that puerarin contained in *Pueraria lobata* (Kudzu) restored autophagy through activation of AMPK and significantly restored LC3B-II and decreased p62 protein content [86]. Moreover, berberine (located on red berries) increased cell viability and autophagy via up-regulation phosphorylation of the insulin receptor and its downstream signaling molecules AMPK, Akt and eNOS in a diabetes rat model [87]. On the other hand, curcumin exerts neuroprotective effects by attenuating autophagic activities through mediating the PI3K/Akt/mTOR pathway, while also suppressing an inflammatory reaction by regulating the TLR4/p38/MAPK pathway on Sprague Dawley male rats [88].

Additionally, retinoic acid has been shown to protect the liver from injury by promoting autophagy, which is dependent on Foxo3/p-Akt/Foxo1 signaling [89]. In this sense, FOXO1 and FOXO3 increase autophagic flux through core ATG gene expression [83]. The relationship between autophagy and apoptosis is represented by the interaction between the anti-apoptotic protein Bcl-2 and the autophagy protein Beclin 1. Interestingly, the use of nutraceutics, such as xanthohumol, baicalin, apigenin, tetrahydropalmatine and emodin, can regulate this complex Bcl-2/Beclin 1 in age-related diseases [90,91,92,93,94,95].

### 3.5. Deregulated Nutrient Sensing

The ability to sense and respond to fluctuations in environmental nutrient levels is fundamental for life. Different pathways that detect intracellular and extracellular levels of sugars, amino acids, lipids and surrogate metabolites are integrated and coordinated at the organismal level through hormonal signals [96]. Nutrient-sensing pathways are commonly deregulated in age-related pathologies, such as stroke, Alzheimer’s disease, breast cancer and liver cancer. According to several authors, the major nutrient-sensing pathways are insulin and insulin/insulin-like growth factor-1 (IGF-1) signaling IIS, mechanistic target of rapamycin (mTOr), AMP-activated protein kinase (AMPK) and sirtuins (SIRTs).

It is noteworthy that in a *C. elegans* model treated with various concentrations of myoinositol (vitamin B8 found in legumes, beans, nuts and other seeds), lifespan increased with attenuation of the IIS pathway in an AKT- and DAF-16-dependent manner. The same study tested the effect of myoinositol on Hs68 cells, resulting in the inhibition of phosphoinositide 3-kinase (PI3K), down-regulation of PI3K expression and inhibition of AKT phosphorylation, which promoted DAF-16 activation; such a mechanism is associated with longevity [97].

In humans, it has been demonstrated that centenarians exhibit specific energetic and metabolic characteristics that contribute to their longevity. For instance, centenarians maintain normal glucose levels and insulin sensitivity [98]. This observation has been associated with a higher plasma proportion of IGF-I/IGFBP-3, indicating greater bioavailability of IGF-1, which means a more effective insulin action response in centenarians [99,100]. 

Another interesting target is mTOR, an evolutionarily conserved nutrient-sensing protein kinase that regulates growth and metabolism in all eukaryotic cells. In AD, the use of a selenium derivative, quercetin and curcumin downregulates Akt/mTOR signaling, induces autophagy and inhibits Aβ generation, respectively [101,102,103]. 

On the other side, AMPK has been identified as a longevity kinase, which can be activated by nutraceuticals, such as resveratrol, genistein, gallic acid and betaine [104]. Genistein (principally found in soybean products) has been tested with vascular smooth muscle cells, where an increase in the phosphorylation of LKB1 and AMPK induces autophagy through the negative regulation of mTOR. Furthermore, genistein exhibits a multimodal mechanism of action, exerting antioxidant, anti-inflammatory, autophagy and senescence-promoting effects; these properties render it a promising nutraceutical candidate in the field of geroscience [105]. 

The activation of SIRTs appears to be a key factor in increasing life expectancy. It has been demonstrated that certain compounds derived from foods, including resveratrol, fisetin and quercetin, can activate SIRT1 [106]. Indeed, in a murine model in which resveratrol was administered, it was observed that SIRT1 activation was associated with AMPK activation and increased levels of NAD(+) in skeletal muscle [107]. AMPK, in turn, contributes to the prolongation of longevity of IIS signaling, thus indicating that these pathways interact in an intricate manner during the process of aging [108]. 

### 3.6. Mitochondrial Dysfunction

Mitochondrial dysfunction during aging is characterized by a loss of efficiency in the electron transport chain and reductions in the synthesis of high-energy molecules, such as ATP [109]. Mitochondrial dysfunction leads to an increased release of reactive oxygen species (ROS) produced during oxidative phosphorylation, causing oxidative damage [110]. In this sense, several efforts to counteract oxidative stress using nutraceuticals have been performed. For instance, isoquercetin and emodin reduce ROS and MDA production while increasing SOD and CAT activity [111,112], while apigenin increased SOD and GSH-Px activities and decreased ROS and MDA levels in a macular degeneration model [113]. 

Moreover, in a recent systematic review, sodium nitrite, PUFA, nicotinamide riboside, urolithin A and whey protein powder improve mitochondrial function in terms of oxidative capacity, antioxidant capacity, mitochondrial volume, bioenergetic capacity and mitochondrial activity, including biogenesis and function [114]. Moreover, another interesting bioactive compound is the α-lipoic acid found in red meat, carrots, beets, spinach, broccoli and potatoes; it is a highly effective antioxidant that inhibits ROS production and improves mitochondrial ATP production [115]. Similarly, it has been reported in 20-month-old male Wistar rats that pea protein combined with inulin (a type of dietary fiber found in various plants) improves mitochondrial activity and biogenesis [116]. Another well-studied polyphenol found in wine is resveratrol, which improves mitochondrial energy metabolism via PGC1α and SIRT1 activation [117]. As well, quercetin, a flavonoid found in fruits and vegetables, has been reported to reduce the prevalence of age-related macular degeneration [118]. This effect could be associated with preventing oxidative stress by ROS scavenging and ameliorates mitochondrial dysfunction via the AMPK/SIRT1 signaling pathways, as has been shown with quercetin [118,119].

### 3.7. Cellular Senescence

Cellular senescence (CS) is an antagonist hallmark of aging defined as a state of irreversible growth arrest and exit from the cell cycle in response to different types of damage (DNA damage, mutations, telomere attrition, ROS and epigenetic disturbances, among others). Senescent cells can secrete a pathogenic senescence-associated secretory phenotype (SASP) that disrupts tissue homeostasis, resulting in loss of tissue repair and the induction of a characteristic inflammatory phenotype associated with aging called *inflammaging*; additionally, under normal conditions and development, SASP could lead to the regeneration signals of tissue [120].

Senescent cells accumulated during aging have been associated with different age-related diseases [121]. CS can be induced by several factors, such as multiple tissue dysfunction, cancer and other pathological conditions related to inflammation. The use of nutraceutics belonging to the flavonoid family (quercetin, fisetin and procyanidin C1) and to the sesquiterpene family (Cis-Nerolidol) have been shown to reduce the expression of genes associated with SASP factors and CCND1, CCNE1, CDK1 and CDK2, respectively [122,123,124,125]. It has been proven that the use of nutraceutical products, such as fisetin and procyanidin C1, have the ability to reduce SASP markers in multiple tissues [123,124]. In recent years, there has been an enormous interest in the development of the so-called senolytics (which is a class of molecules that selectively induce the death of senescent cells) [126]. Interestingly, among the most studied senolytic drugs are dasatinib, quercetin, fisetin and navitoclax. In particular, quercetin and fisetin have been isolated from onions, apples, grapes, berries, broccoli, citrus fruits, cherries, green tea, coffee, red wine, capers, strawberries, apples, grapes and cucumbers. Even so, recently, luteolin contained in peppers, parsley, celery and broccoli was implicated in the suppression of the SASP factor [127]. Moreover, the combination of dasatinib and quercetin has been shown to act as a potent senolytic that ameliorates numerous age-related disorders related to intestinal senescence and inflammation through SASP markers reduction [125].

### 3.8. Stem Cell Exhaustion

Stem cells have the potential to give rise to all tissue types, serving as a source of cells for repairing tissues in the organisms; however, as we become older, tissues experience a progressive decline in homeostatic and regenerative capacities that have been associated with degenerative deleterious alterations in systemic cues and niches that are involved in both the stem cell activity and quantity [128]. In this context, stem cells have a unique metabolism and nutrient needs as compared with other differentiated cell types. 

Interestingly, it has been reported that both stem cell quantity and quality are influenced by diets and dietary patterns [129]. For instance, the intestinal stem cells (Lgr5+ and 4+) are stimulated by ketogenic diets leading to the generation of ketones in mitochondria by the 3-hydroxy-3-methylglutaryl-CoA synthase 2. However, the inhibition of this impairs stemness in Lgr5+ stem cells [130]. Interestingly, it has been reported that an oral nutritional supplement containing resveratrol, vitamin D3 and inositol hexaphosphate (Longevinex^®^, Las Vegas, NV, USA) maintains stem cell survival of retinal pigment epithelium in older adults. Additionally, authors report beneficial effects on structure and visual function in living human eyes [131].

### 3.9. Altered Intercellular Communication

Aging has been associated with progressive alterations in cellular communication that comprise cellular signaling, leading to cellular pathological states. There are several examples of such in different diseases, for instance, carcinogenesis, neurodegeneration and cardiovascular hypertrophy. In this regard, baicalin has a significant neuroprotective effect in stroke by increasing GABA(A)R α1, GABA(A)R γ2 and KCC2 mRNA and protein levels, facilitating neurological function and suppressing ischemia-induced neuronal damage [132]. On the other hand, vitamin D has shown to be an interesting immunomodulatory molecule that could interact with immune cells [133]. Another interesting compound is epicatechin present in cocoa powder, which induces SOD activity, IGF-1 pathways and mitochondrial biogenesis [133,134]. Additionally, curcumin and ginsenosides induce neuroprotective effects by increasing the expression of both *Creb* and *Bdnf* genes and concomitantly leading to reduced apoptosis [135,136,137].

### 3.10. Chronic Inflammation (Inflammaging)

Most older individuals develop “*inflammaging*”, a condition characterized by elevated levels of blood inflammatory markers that carries high susceptibility to chronic morbidity, disability, frailty and premature death [138]. In recent years, increasing evidence suggests that nutrient interventions (natural or synthetic) have an influence in delaying and preventing *inflammaging*. For instance, resveratrol present in several nutraceutics, grapes or fruits, has been shown to inhibit NF-κB-regulated cytokines; data suggest that resveratrol possesses a significant effect against inflammation through the SIRT1 pathway [139]. Quercetin is a pentahydroxyflavone found on several fruits and vegetables, including onions, capers, apples, berries, tea, tomatoes, grapes, brassica vegetables and shallots. It is also found in various nuts, seeds, barks, flowers and leaves [140,141]. Quercetin has been shown to be an inhibitory compound of cyclooxygenase and lipoxygenase enzymes, both quite crucial in the mediation of prostaglandins and leukotrienes in inflammation [141,142,143,144]. Additionally, it was reported that quercetin could reduce the secretion of anti-inflammatory cytokines, such as IL-10, and pro-inflammatory cytokines, such as TNF-α, IL-1B and IL-6 [145]. Furthermore, we recently found that quercetin blocks airway smooth muscle contraction by inhibiting L-VDCC and SOCC [146]. Another interesting compound is epigallocatechin-3-gallate, which is present in several nutritional supplements and is the main component of green tea; such compounds have shown to exert anti-inflammatory effects since it can inhibit (PI3K)/Akt/mTOR pathway quite related to aging [147].

Other nutraceuticals from the families of bioactive compounds, such as flavonoids (baicalin and carthamin yellow), flavonols (icariin), flavones (apigenin and scutellarin), alkaloids (berberine), stilbenoids (resveratrol) and adenosine analogs (cordycepin), have been associated with a reduction in blood inflammatory markers (TNF-α, IL-1β, IL-6, IL-18, TGF-β1 and TGF-β2) in in vivo and in vitro models [148,149,150,151,152,153,154]. Moreover, the use of γ-tocotrienol repressed inflammasome activation, caspase-1 cleavage and interleukin (IL) 1β secretion in murine macrophages, implicating NLRP3 inflammasome inhibition, thereby delaying the progression of type 2 diabetes [155]. 

### 3.11. Dysbiosis

In recent years, many of the modern multifactorial diseases, such as neurodegenerative and metabolic disorders, have shown increasing evidence of an abnormal microbiome structure (dysbiosis), which affects the taxonomic composition, as well as the metagenomic function of the microbial community [156]. Moreover, studies on centenarians showed a youth-associated gut microbiome characterized by an over-representation of a Bacteroides-dominated enterotype [157] and a quite diverse gut virome, including a virus associated with Clostridia [158]. The gut microbiota can metabolize compounds from nutraceuticals to produce new absorbable small molecules, which have active effects. Also, nutraceuticals can regulate the composition of gut microbiota and its secretions, and such secretions may play a therapeutic role [159]. For instance, several flavonoids, polysaccharides and saponins can promote the growth of beneficial gut microbiota; a good example of such is the metabolism of ginsenosides, which are susceptible to Lactobacillus, Bacteroides, Bifidobacterium and their metabolic enzymes, which give rise to secondary ginsenosides with different pharmacological properties (antiangiogenic and anticancer) and are more easily absorbable in the circulation [159].

Another interesting result is the use of a mixture of quercetin (highly present in nutraceuticals and known to have senolytic potential) with dasatinib, which was shown to induce changes in the intestinal microbiota, specifically in the ileum of treated mice versus controls. These changes are associated with (1) a slightly lower Firmicutes-Bacteroidetes ratio, which correlated negatively with inflammatory and senescence markers in all the intestinal sections and (2) increased Akkermansia abundance, which has been linked with reduced intestinal permeability and gut-to-blood leakage of endotoxins, thereby alleviating diet-induced obesity and insulin resistance [125]. 

### 3.12. Genomic Instability

Genomic instability is a strong contender as a major cause of aging. The accumulation of DNA damage and mutations within a cell’s genome over time causes a remarkable instability in the DNA sequence, leading to indels, chromosomal abnormalities, substitutions, breaks, gaps and aberrant 3-D structures, among others. Interestingly, genomic instability is associated with all the hallmarks of aging, since in the end all lead to DNA damage, such as ROS or RNS. On the other side, genomic instability has been associated with the development of several age-related diseases, such as cancer or neurodegeneration, as well as with mortality [160]; therefore, such a process is quite important in geroprotectors development. In this context, it has been suggested that exogenous antioxidants derived from food, such as fruits, vegetables, cereals, mushrooms, nuts and spices, aid in ameliorating the DNA damage caused by oxidative stress triggered by aging per se or by the exposure to xenobiotics, such as of cigarette smoking, alcohol, radiation or environmental toxins [161]. For instance, anthocyanins from grape juice decrease the oxidative damage in cells [162]. Also, polyphenols and melanoidins obtained from coffee brews (caffeoylquinic acids, feruloylquinic acids, dicaffeoylquinic acids and p-coumaroylquinic acids) augmented active oxygen-scavenging activity [163]. Moreover, flavonoids are a set of compounds contained in several fruits with quite powerful antioxidant properties useful against ROS. On the other hand, resveratrol isolated from *Polygonum cuspidatum* but also contained in several nutraceuticals stimulates SIRT2 activity, increasing DNA stability in yeast of *S. cerevisiae* [164]. Another interesting set of compounds contained in nutraceuticals are curcumins derived from *Curcuma longa*. They decrease lipid peroxidation and β-carotene inducing quenching activities associated with protecting biological systems from ROS-mediated damage [165].

## 4. State of the Art on Nutraceuticals Research

As we describe in Table 2, it has been reported that nutraceuticals target at least one of the twelve hallmarks of aging. In many cases, such analyses were performed experimentally; however, this represents high costs, and depending on the experimental model, the results are quite heterogeneous, representing a limitation for translational medicine. In this sense, computational techniques (chemoinformatics), such as network pharmacology, molecular modeling, quantitative or qualitative “structure-activity” relationships at different quantic levels (QSAR 2-D, 3-D, 4-D) and artificial intelligence methods, have been widely used in the discovery of potential nutraceuticals that may be applied as geroprotectors [166]. Such approaches lead to overpassing the limitations of experimental strategies and concomitantly accelerating and optimizing drug discovery. Moreover, in recent years the field of Artificial Intelligence technologies, including machine-learning methods, such as deep-learning, SVM and cognitive computing, have stimulated the process of chemical data processing obtained from omics data and clinical trials, leading to obtaining novel solutions to achieve healthy aging or to repurpose compounds to be used against age-related diseases. Although such a topic is out of the scope of the present review, we encourage our readers to refer to [167,168]. 

## 5. Concluding Remarks State of the Art on Nutraceuticals Research

The concept of food as medicine has been around for centuries, with Hippocrates famously stating, “*Let food be thy medicine and medicine be thy food*”. As we age, our bodies become more susceptible to disease and decline in function. This has led to increased interest in strategies like consuming nutraceuticals to promote healthy aging. This review explores the potential of nutraceuticals as geroprotectors, substances that may delay or prevent age-related diseases. We highlight that not all nutraceuticals qualify as geroprotectors. Only those that target multiple hallmarks of aging, as defined by criteria established by researchers like Moskalev et al., have this potential. Vitamin D, curcumin, resveratrol, quercetin, genistein, gallic acid, baicalin, lipoic acid and epicatechin are already reported as geroprotectors at the geroproctector.org [200]. In this context, several nutraceuticals, including vitamins A, C and E, EPA, DHA, daidzein and aloin, among others, may be considered promising geroprotectors that require further research, including clinical trials and studies investigating the molecular mechanisms of these nutraceuticals. Moreover, understanding their optimal dosage and long-term safety in humans is also crucial. Future studies should also explore the potential benefits of combining different nutraceuticals and how they interact with lifestyle factors like exercise. By continuing research in this field, we may develop strategies to improve health span and promote longevity.

## Figures and Tables

**Figure 1 nutrients-16-02835-f001:**
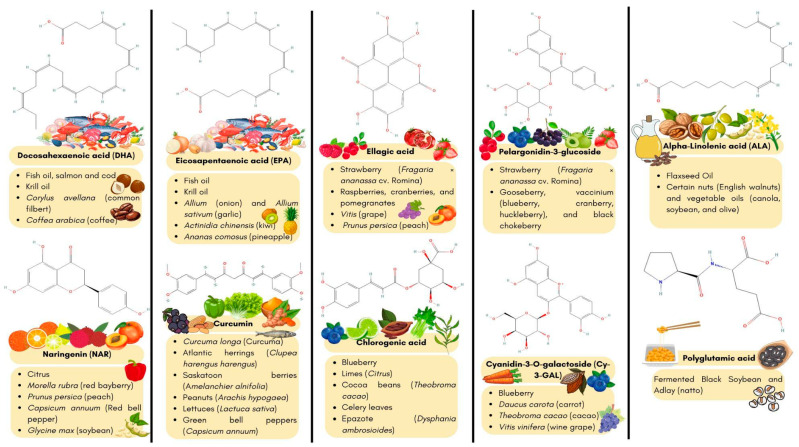
Examples of the enriched sources of the most common nutraceuticals. This figure depicted the most common nutraceuticals found in vegetables, roots, fruits, nuts and seafood. For instance, green bell peppers, lettuce, celery and epazote are enriched in curcumin and chlorogenic acid. However, red bell, peach, soybean, grapes, strawberries and other berries are enriched sources of naringenin, ellagic acid, pelargonidin-3-glucoside and cy-3-gal. Seafood, nuts and vegetable oils are sources enriched in fatty acids, such as DHA, EPA and ALA.

**Table 1 nutrients-16-02835-t001:** General overview of experimental evidence on selected nutraceuticals with potential use in aging or age-related diseases.

Nutraceutical Source	Bioactive Compounds or Organisms	Nutraceutical Classification	Age-Related Target (Pharmacological or Biological Activities)	Refs.
Wheatgrass (*Triticum aestivum*)	Chlorophyll Flavonoids Vitamin C Vitamin E	Phytochemicals Antioxidants Vitamins	Decreases triglycerides in blood. Inhibits growth of leukemia cells. Benefits immunological activity. Decreases oxidative stress.	[17,18,19]
*Aloe vera* extract	Quercetin Myricetin Aloin Vanillic acid Palmitic acid Vitamin E Polysaccharides Phenolic compounds	Phytochemicals Antioxidants	Aloe is useful for photoaging since it stimulates fibroblast, which produces collagen and elastin fibers, making the skin more elastic and less wrinkled. Additionally, it inhibits the cyclooxygenase pathway and reduces prostaglandin E2 production from arachidonic acid. Quercetin, which exists in the outer layers of aloe leaf, has a cytoprotective effect on mitochondrial pathways by inhibiting oxidative stress.	[20,21,22]
Ginseng extract *Panax ginseng*	Ginsenoside C-K Oleanolic acid Ginsenoside Rg1-Rb1, Rd Oleanane Polysaccharides Peptides Phenolic compounds	Phytochemicals	Ginseng exhibited a remarkable antioxidant effect through the enhancement of the cell stress response, mainly by up-regulating heme oxygenase-1. In a rat model of high-fructose diet-induced metabolic disorder, fermented red ginseng reduced hyperlipidemia and hypertension. An aqueous extract of Korean red ginseng rapidly up-regulated endothelial NO synthase (eNOS) via the phosphoinositide 3-kinase (PI3K)/Akt-pathway in human umbilical vein endothelial cells (HUVEC).	[23,24,25]
Seaweed species *Hypnea musiformis*, *Ochtodes secundiramea*, *Padina gymnospora*, *Codium tomentosum*, *and Pterocladiella capillacea*	Fucoidan Fucoxanthin Phycoerythin Alginic acid Polysaccharides Carotenoids Taurine	Phytochemicals Amino acid	Seaweed is reported to ameliorate or prevent Aβ_25–35_ aggregation and inhibit AChE and BuChE levels in vitro. MeOH extracts of seaweed *S. muticum* and *S. polyschides* exhibited the highest neuroprotective effects against dopamine-induced neurotoxicity in SH-SY5Y cells.	[26,27]
*Echinacea purpurea* extracts	Caffeic acid β-sitosterol Phenolic compounds	Phytochemicals	After 8 weeks of *Echinacea consumption*, a significant increase in NK cell cytotoxic activity was observed. Serum cytokine levels of IL-2, IFN-γ, and TNF-α also significantly increased. In vitro gastrointestinal digestion on the phenolic composition of Echinacea extracts showed significant reductions in IL-6, IL-8, and PGE2 levels in vitro.	[28,29]
Goji berry (*Lycium barbarum)* extract	L. barbarum polysaccharides (LBPs) Pectic polysaccharides Lycopene Beta-carotene Lutein Zeaxanthin Phenolic compounds Rutin	Phytochemicals Antioxidants	Improve mitochondrial function and decrease oxidative stress via Nrf2-Maf and NOS signaling pathways. Improve cognitive performance in aged rats by decreased astrogliosis.	[30]
Chiang-Da (*Gymnema inodorum*) leaf extracts	(3β, 16β)-16,28-dihydroxyolean-12-en-3-yl-*O*-β-d-glucopyranosyl-β-d-glucopyranosiduronic acid (GIA1)	Phytochemicals	Induces anti-hyperglycemic mechanisms by reducing α-glucosidase activity and glucose transport of SGLT.	[31]
Strawberry *(Fragaria x ananassa cv. Romina)* extracts	Ellagic acid Pelargonidin-3-glucoside (Phenolic compounds) K^+^, Mg^+^, P^+^ and Ca^2+^ (Minerals)	Phytochemicals Essential trace elements	Induces DAF-16/FOXO and SKN-1/NRF2 pathways. Delay β-amyloid induced paralysis Reduced β-amyloid aggregation Prevents oxidative stress in *C. elegans.*	[32]
Fish hydrolysate	Eicosapentaenoic acid (EPA) Docosahexaenoic acid (DHA)	Fatty acids	Improved memory performance in aged mice. Regulates gut microbiota. Regulates corticosterone levels. Increased the expression of the mitochondrial respiratory chain (ND1, ND2, ND5, and ND6). Improving total skeletal muscle mass, muscle strength and physical performance in older adults.	[33,34]
Blueberry (*Vaccinium uliginosum* L.) extracts	Polyphenolic compounds Cyaniding-3-O-galactoside (Antocyanin) Pyruvic acid Chlorogenic acid	Phytochemicals	Promotes recovery from cell injury and improves survival of hippocampal pyramidal neurons. Increases antioxidant defenses via ERK signaling pathway in the hippocampus of a senescence-accelerated mouse model.	[35,36]
Tempeh (soybean fermentation)	Daidzein Genistein Polyphenols Low-molecular-weight soluble dietary fiber Tempeh isoflavone Peptides: Ala-Val, Gly-Leu, Gly-Phe, Pro-Leu, Ala-Phe, Asp-Met, Asp-Tyr, Pro-Ala-Pro, Ile-Ala-Lys, Arg-Ile-Tyr and Val-Ile-Lys-Pro.	Phytochemicals Dietary fiber Antioxidants Proteins and amino acids	Induces Anti-inflammatory and immunomodulatory components. Improve antioxidative activity and increase both SOD and CAT gene expression. Induces anti-hypertensive activity via ACE inhibitor peptide Induces neuroprotection and GABA synthesis in six-month-old senescence-accelerated mice.	[37,38,39,40]
Curcumin C3 complex	Polyphenolic orange-yellow pigments: curcumin, demethoxycurcuminbis-demethoxycurcumin	Phytochemicals	Decreased IL-6 concentration and gene expression. Prevents senescent cell accumulation. Improve antioxidant capacity. Upregulate TERT gene expression Increased telomere length in aged rats. Upregulate TERT gene expression Increased telomere length in aged rats (17 months old).	[41]
Blueberry (*Vaccinium uliginosum* L.) extract	Flavonoids (anthocyanidins) Polyphenols (procyanidin) Phenolic acid Pyruvic acid Chlorogenic acid			
*Astragalus membranaceus*	Astragaloside IV Kaempferol Quercetin Isorhamnetin Triterpene saponins			
*Amelanchier ovalis* berries ethanolic extract	Gallic acid p-hydroxybenzoic acid Protocatechinic acid	Phytochemicals	Promotes proliferation, lifespan and survival rate of *Saccharomyces cerevisiae* Y-564 exposed to oxidative stress.	[42]
Krill oil	Astaxanthin Choline Omega-3 DHA EPA	Phytochemicals Vitamin precursors Fatty acids	mTOR-p70s6k/Muscular strength and cognitive function The administration of krill oil to a mixed-sex aged C57BL/6 mouse model increased force production (increased grip strength, increased contraction and tetanic strength in the extensor digitorum longus muscle) without altering Ca^2+^ homeostasis in the excitation-contraction coupling mechanism or mitochondrial Ca^2+^ uptake processes.	[43]
*Lycium ruthenicum Murr* ethanolic extract	Anthocyanins Lycibarbar spermidine B N1-Dihydrocaffeoyl N10-trans-caffeoyl-spermidine)	Phytochemicals	Prevents oxidative damage by increasing SOD and glutathione peroxidase concentration in a murine model of accelerated aging induced by D-galactose.	[44]
Fermented Black Soybean and Adlay (FBA)	Nattokinase Polyglutamic acid Isoflavones	Proteins and amino acids Phytochemicals	Improves body composition in aged mice (increased gastrocnemius muscle and decreased fat accumulation). Interestingly, it reduced the expression of GLB1 and p16^INK4A^ genes involved in senescence. Counteracts oxidative stress. Decrease inflammation markers MCP-1, IL-6 and IL-10 in aged mice. Improves aging-related gut microbial dysbiosis promoting the growth of beneficial microbes (Alistipes, Anaeroplasma, Coriobacteriaceae UCG002, and Parvibacter).	[45]
Soybean	Daidzein Genistein Glycitein Acetyldaidzin Acetylgenistin Acetylglycitin	Phytochemicals	Induces anti-photoaging in murine models exposed to UVB radiation.	[46]
Fermented milk	*Lactobacillus paracasei* *Lactobacillus plantarum*	Prebiotics or Probiotics	Improve symptoms associated with allergic rhinitis. Reduces airway hyperresponsiveness, asthma and systemic proinflammatory factors (IL-4, IL-5, and IL-3).	[47,48]

**Table 2 nutrients-16-02835-t002:** Nutraceuticals and the hallmarks of aging. Of particular interest are bioactive compounds contained in nutraceuticals, such as vitamins A, D, and E, as well as curcumin, resveratrol, quercetin, apigenin, genistein, and fatty acids such as EPA and DHA, which have been shown to act on at least three hallmarks of aging, indicating potential as geroprotectors. Telomere Attrition (T.A.), Epigenetic Mechanisms (EP.), Loss of Proteostasis (L.P.), Disabled Macroautophagy (D.M.), Deregulated Nutrient-Sensing (D.N.S.), Mitochondrial dysfunction (M.D.), Cellular Senescence (C.S.), Stem Cell Exhaustion (S.C.E.), Altered Intercellular Communication (A.I.C.), Chronic Inflammation (C.I.), Dysbiosis (Dys.) and Genomic Instability (G.I.). (✔) Targeted hallmarks by nutraceuticals.

Nutraceutical Source	Bioactive Compounds	Hallmarks of Aging												Refs.
		T.A.	EP.	L.P.	D.M.	D.N.S.	M.D.	C.S.	S.C.E.	A.I.C.	C.I.	Dys.	G.I.	
Dairy products, Cod liver oil, Fish oil, Beef liver, Carrot seed oil, Palm fruit oil	Vitamin A (retinoic acid)	✔	✔		✔									[53,68,89]
Wheatgrass, Acerola cherry extract, Rosehip extract, Camu camu extract, Sea buckthorn oil	Vitamin C	✔	✔											[52,54,63,69]
Cod liver oil, Fish oil, Lanolin	Vitamin D	✔	✔						✔	✔				[52,54,69,131,133]
Wheat seed oil, Sunflower oil, Almond oil, Soybean oil, Acai berry extract	Vitamin E	✔	✔								✔			[54,63,155]
Turmeric extract, Curcumin C3 complex,	Curcumin	✔	✔	✔	✔	✔				✔			✔	[51,63,70,74,75,85,88,102,135]
Grape seed extract Red wine extract Blueberry extract Cranberry extract Peanut extract	Resveratrol	✔	✔	✔		✔	✔		✔		✔		✔	[51,57,63,69,70,81,104,106,107,117,131,139,164]
*Aloe vera* extract Quercetin supplements Multi-antioxidant formulas Onion extract Apple extract Broccoli extract	Quercetin				✔	✔	✔	✔			✔	✔		[84,103,106,118,125,127,145,169]
Fish oil Krill oil Seaweed oil	EPA	✔	✔				✔							[55,58,69,114]
Fish oil Krill oil Seaweed oil	DHA	✔	✔				✔							[55,58,69,114]
Sunflower oil Corn oil Soybean oil Grape seed oil Hemp seed oil	Linoleic Acid	✔		✔										[58,74]
Chamomile extract Parsley extract Celery seed extract Citrus bioflavonoid complex	Apigenin				✔		✔				✔			[93,113]
Tempeh Red clover extract Soybean Soy isoflavone supplements	Genistein				✔	✔			✔		✔			[104,105]
*Amelanchier ovalis* berries ethanolic extract Pomegranate extract Green tea extract Grape seed extract Acai berry extract	Gallic acid					✔								[104]
Wheat germ Quinoa	Betaine					✔								[104]
Grape seed extract Pine bark extract Cocoa extracct	Procyanidin C1							✔						[123]
Carrots, peppers, thyme, broccoli, onion leaves, cabbages, apple skins, rosemary, parsley, and spinach	Luteolin							✔						[127]
Krill oil Seaweed-based supplemnts	Astaxanthin		✔											[69]
Olive leaf extract Olive oil	Oleuropein			✔										[80]
*Scutellaria baicalensis* root extract	Baicalin				✔					✔	✔			[94,132,149]
Multivitamin/mineral supplements α-lipoic acid supplements	α-lipoic acid						✔							[115]
Inulin supplements Prebiotic supplements Probiotic and prebiotic combination Chicory root extract	Inulin						✔							[116]
Cocoa extract Dark Chocolate Green tea extract	Epicatechin									✔				[134]
Soy isoflavone supplements Soy-based products	Daidzein	✔	✔				✔			✔	✔			[170,171,172,173]
Seaweed oil Tomato extract Palm fruit oil Carrot seed oil Mixed carotenoid supplements	Carotenoids	✔			✔		✔	✔			✔		✔	[174,175,176,177]
Brown algae extract Seaweed-based supplements	Fucoxanthin				✔						✔			[178,179,180]
Brown algae extract Seaweed-based supplements	Fucoidan				✔			✔		✔	✔			[181,182,183,184,185]
Ginseng root extract Ginseng containing supplements	Ginsenosides C-K				✔						✔			[186,187,188]
Korean Red Ginseng extract American Ginseng extract	Ginsenosides Rg1-Rb1, Rd	✔					✔	✔	✔		✔			[189,190,191,192,193]
*Aloe vera* gel *Aloe vera* extract	Aloin		✔	✔	✔				✔	✔	✔			[194,195,196,197,198,199]

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
