# Peer review of "Exploring the Geroprotective Potential of Nutraceuticals"

_nutrients, 2024, doi:10.3390/nu16172835_

Round 1

Reviewer 1 Report

Comments and Suggestions for Authors

The manuscript entitled "Can nutraceuticals be considered as geroprotectors?" is a review of nutraceuticals as geroprotectors. The topic is interesting and could be useful for a wide audience of researchers. However, there are some points to increase the visibility of the manuscript.

1 - Abstract and introduction section

Please add the information on the covered years, the number of papers selected, and the select criteria.

2 - Please, I strongly suggest adding a subsection of state of art methodologies used to study nutraceuticals as geroprotectors that can be connected with the conclusion section/remarks about future studies.

Author Response

The manuscript entitled "Can nutraceuticals be considered as geroprotectors?" is a review of
nutraceuticals as geroprotectors. The topic is interesting and could be useful for a wide audience
of researchers. However, there are some points to increase the visibility of the manuscript.

Q1 - Abstract and introduction section. Please add the information on the covered years, the
number of papers selected, and the select criteria.

R: Thanks for your suggestion. Accordingly, in the reviewed version of the manuscript, we have
included a section that describes the covered years, the number of papers selected, and the
selection criteria. Please refer to the main manuscript for more details.

Q2 - Please, I strongly suggest adding a subsection of state of art methodologies used to study
nutraceuticals as geroprotectors that can be connected with the conclusion section/remarks
about future studies.

R: Thank you for your suggestion. We have included a brief subsection describing the role of
computational techniques currently used for such purposes. Please refer to the main manuscript
for further details.

Reviewer 2 Report

Comments and Suggestions for Authors

Based on the title of the study, I expected a slightly different review. In the introduction, I suggest the authors develop information about nutroceuticals. In the second point, the authors included in the table several nutraceuticals that are the best known. I'm afraid they're wrong. There are plenty of well-known nutroceuticals. Therefore, my question is: Why did you choose these nutraceuticals presented in the table 1? I would also expect a more detailed description of them. The "Model" column in Table 1 is not clear to me. If the authors want towrite that nutroceuticals can be consider as geoprotectors, maybe it would be good if they showed better research on the role of nutroceuticals in this aeria. Can all nutraceuticals be classified as geoprotectors? If not, why?

Author Response

Based on the title of the study, I expected a slightly different review. 

Q1 -  In the introduction, I suggest the authors develop information about nutroceuticals.

R1: Thank you for your suggestion. In the reviewed version of the manuscript, we have added information on nutraceuticals, the global market, and the potential uses of such compounds. Moreover, we improved Table 1 for a better understanding. Please refer to the main manuscript for further details. 

Q2 - In the second point, the authors included in the table several nutraceuticals that are the best known. I'm afraid they're wrong. There are plenty of well-known nutroceuticals. Therefore, my question is: Why did you choose these nutraceuticals presented in the table 1? I would also expect a more detailed description of them. 

R2: Thanks for your commentary; we have improved Table 1 according to reviewers' suggestions to be more detailed and include the most studied nutraceuticals in such a section.  Table 1 is a general overview of nutraceuticals with potential against ageing selected from a review of 69 articles that fulfil our inclusion criteria.

Q3 - The "Model" column in Table 1 is not clear to me. 

R3: Thank you for your comment. We have modified Table 1 to fit your suggestions. Please refer to the main text for further details.

Q4 - If the authors want to write that nutroceuticals can be consider as geoprotectors, maybe it would be good if they showed better research on the role of nutroceuticals in this aeria.

R4: Thanks for your kind opinion. We modified Table 1 and added a Table describing the hallmarks of aging that many nutraceuticals target and their justification as potential geroprotectors. 

Q5.Can all nutraceuticals be classified as geoprotectors? If not, why?

R5: This is a very interesting question; we included a paragraph in the conclusion which suggests that a not all nutraceuticals can be classified as geroprotectors only those which meet the criteria established by Moskalev et al. (An intervention that delays, reduces and/or prevents diseases associated with aging and that is characterized by having as a simultaneous target one or several of the pillars of aging). Please refer to the Conclusion section for more details.

Reviewer 3 Report

Comments and Suggestions for Authors

The authors tried to summarize the action of nutraceuticals as geroprotectors and their possible potential use for developing a geroprotective diet. The authors definitions for a nutraceutical is: Biological active molecules naturally occurring in foods that, besides having a nutritional role, provide health-promoting, disease-curing, or prevention properties. A nutraceutical must be understood as a single substance that may be isolated for clinical purposes or consumed as part of a specific food.                     

  Nutraceuticals,                                                                                             In Table1 there are many foods which don’t fit the nutraceutical definition. The table should be changed.                           

Epigenetics," it has been demonstrated that polyphenols, flavonoids, and organosulfur compounds in foods (vitamin A, vitamin C, vitamin E, curcumin, and resveratrol) exert epinutraceutical effects, modifying DNA methylation patterns, histone modifications, and miRNA expression [44]". Too general description without explaining how they acts.                                                                                                   Line-167 Is not in contrast, all those nutraceuticals are modulators of DNA methylation and inhibitors of DNMT.

Line- How quercetin act -no reference, however reference 95 is about orange and hesperidin!!!!

Quercetin, which is a pentahydroxyflavone found on several fruits, mainly citrus, has shown to be an inhibitory compound of cyclooxygenase and lipoxygenase enzymes, both quite crucial in the mediation of prostaglandins and leukotrienes in inflammation [116].  Article 116 have nothing to do with lipoxygenase and cyclooxygenase. Citrus did not contain quercetin in significant amount exactly with tea which did not contain quercetin in significant amounts.

This article contains too many inaccuracies!!!.    

Author Response

Reviewer 3
The authors tried to summarize the action of nutraceuticals as geroprotectors and their potential
use for developing a geroprotective diet. The authors definitions for a nutraceutical is: Biological
active molecules naturally occurring in foods that, besides having a nutritional role, provide
health-promoting, disease-curing, or prevention properties. A nutraceutical must be understood
as a single substance that may be isolated for clinical purposes or consumed as part of a specific
food.

Q1 - Nutraceuticals: Table 1 includes many foods that don’t fit the nutraceutical definition. The
table should be changed.

R1: Thank you for your comment. We have updated Table 1 according to your suggestions. Please
refer to the main text for further details.

Q2 - Epigenetics," it has been demonstrated that polyphenols, flavonoids, and organosulfur
compounds in foods (vitamin A, vitamin C, vitamin E, curcumin, and resveratrol) exert
epinutraceutical effects, modifying DNA methylation patterns, histone modifications, and miRNA
expression [44]". Too general description without explaining how they acts.

R2: Thanks for your commentary, we have added information on the most common mechanism by
which compounds act. Nevertheless, it should be mentioned that the exact effect mechanisms by
which vitamins interact with several epigenetic mechanisms is a process that is yet to be
investigated.

Q3 - Line-167 Is not in contrast, all those nutraceuticals are modulators of DNA methylation and
inhibitors of DNMT.

R3: We have corrected the redaction; please refer to the main manuscript.

Q4 - Line- How quercetin act -no reference, however reference 95 is about orange and
hesperidin!!!!

R4: Thank you for your feedback. We have made the necessary corrections to the article and
included a reference. Additionally, reference 95 has been revised. The author of the quote states
that "involved key individual flavonoids-quercetin (a flavonol) and hesperidin (flavanone)-and the
prevalence of age-related macular degeneration (AMD)" In the results section, the authors state
that "Quercetin was associated with reduced odds of any AMD (OR: 0.76; 95% CI: 0.58, 0.99)"
(Gopinath et al., 2018).

Gopinath B, Liew G, Kifley A, Flood VM, Joachim N, Lewis JR, Hodgson JM, Mitchell P. Dietary
flavonoids and the prevalence and 15-y incidence of age-related macular degeneration. Am J Clin
Nutr. 2018 Aug 1;108(2):381-387. doi: 10.1093/ajcn/nqy114. PMID: 29982448.

Q5 Quercetin, which is a pentahydroxyflavone found on several fruits, mainly citrus, has shown to
be an inhibitory compound of cyclooxygenase and lipoxygenase enzymes, both quite crucial in
the mediation of prostaglandins and leukotrienes in inflammation [116]. Article 116 have nothing
to do with lipoxygenase and cyclooxygenase. Citrus did not contain quercetin in significant
amount exactly with tea which did not contain quercetin in significant amounts.

R5: Thank you for your observation. Accordingly, we recognized that quercetin is an important
plant metabolite, naturally present in many fruits and vegetables, including onions, capers,
apples, berries, tea, tomatoes, grapes, brassica vegetables, and shallots. It is also found in nuts,
seeds, barks, flowers, and leaves (Hollman et al., 2000; Mlcek et al., 2016). Additionally, quercetin
is an inhibitory compound of cyclooxygenase and lipoxygenase enzymes, both crucial in
mediating prostaglandins and leukotrienes in inflammation (Kim et al., 1998; Lee et al., 2010).
Furthermore, we recently found that quercetin blocks airway smooth muscle contraction by
inhibiting L-VDCC and SOCC (Flores-Soto et al., 2024). In this sense, we have corrected and
updated the references in the reviewed version of the manuscript according to our purposes.
Please refer to the main manuscript for more details.

Flores-Soto E, Romero-Martínez BS, Solís-Chagoyán H, Estrella-Parra EA, Avila-Acevedo JG,
Gomez-Verjan JC, Reyes-García J, Casas-Hernández MF, Sommer B, Montaño LM. Chamaecyparis
lawsoniana and Its Active Compound Quercetin as Ca2+ Inhibitors in the Contraction of Airway
Smooth Muscle. Molecules. 2024 May 12;29(10):2284. doi: 10.3390/molecules29102284. PMID:
38792145; PMCID: PMC11123793.

Hollman, P. C.; Arts, I. C. Flavonols, flavones and flavanols - nature, occurrence and dietary
burden. J. Sci. Food Agric 2000, 80(7), 1081–1093.
doi:10.1002/(sici)1097-0010(20000515)80:7<1081::aid-jsfa566>3.0.co;2-g
Kim H.P., Mani I., Iversen L., Ziboh V.A. Effects of naturally-occurring flavonoids and bioflavonoids
on epidermal cyclooxygenase and lipoxygenase from guinea-pigs. Prostaglandins Leukot. Essent.
Fat. Acids. 1998;58:17–24. doi: 10.1016/S0952-3278(98)90125-9.
Lee K.M., Hwang M.K., Lee D.E., Lee K.W., Lee H.J. Protective effect of quercetin against
arsenite-induced COX-2 expression by targeting PI3K in rat liver epithelial cells. J. Agric. Food
Chem. 2010;58:5815–5820. doi: 10.1021/jf903698s.
Mlcek, J.; Jurikova, T.; Skrovankova, S.; Sochor, J. Quercetin and Its Anti-allergic Immune
Response. Molecules 2016, 21, 623, doi:10.3390/molecules21050623.

Q6 This article contains too many inaccuracies!!!.

R: Thanks for your kind opinion. We have corrected the article according to your suggestions and
the other reviewers' suggestions. Please refer to the main manuscript.

Round 2

Reviewer 2 Report

Comments and Suggestions for Authors

The manuscript can be published in Nutrients

Author Response

Comment: The manuscript can be published in Nutrients. 

Response: Thanks for your kind review and commentaries. We are sure that they improved the quality of our article.